# Does Quality of Care (QoC) Perception Influence the Quality of Life (QoL) in Women with Endometriosis? Results from an Italian Nationwide Survey during Covid Pandemic

**DOI:** 10.3390/ijerph20010625

**Published:** 2022-12-29

**Authors:** Vincenza Cofini, Mario Muselli, Chiara Lolli, Leila Fabiani, Stefano Necozione

**Affiliations:** Department of Life, Health and Environmental Sciences, University of L’Aquila, 67100 L’Aquila, Italy

**Keywords:** quality of life, endometriosis, womens health, quality of healthcare, mental health, physical health, prevention, management, public health

## Abstract

(1) Background: Endometriosis is a chronic and progressive illness that generates a slew of issues, lowering the quality of life of women. The purpose of this study was to look at the quality of life in women with endometriosis and how it relates to the quality of care. (2) Methods: This study is an online survey performed in Italy during the COVID pandemic using the Italian version of the Health Questionnaire SF-36 and a questionnaire for assessing the quality of care received. (3) Results: 1052 women with a self-reported diagnosis of endometriosis participated in the survey. The mean levels of Physical Component Summary (PCS) and Mental Component Summary (MCS) were 38.89 ± 10.55 and 34.59 ± 11.17, respectively. A total of 77% of women judged the services they received positively, and 51% considered the coordination between healthcare professionals to be satisfactory. The satisfaction index mean was 23.11 ± 4.80. PCS was positively related to Occupation, high educational level, physical activity, and health care satisfaction. MCS was positively related to higher age, physical activity, and health care satisfaction. (4) Conclusions: The study indicated that satisfaction with health care was a significant predictor of QoL in women with endometriosis, for both physical and mental health.

## 1. Introduction

Endometriosis is a disease characterized by the presence of tissue resembling the endometrium outside of the uterus [1].

As reported by the World Health Organization (WHO), endometriosis is a chronic and disabling disease of reproductive age, leading to pelvic discomfort and infertility because it creates a persistent inflammatory reaction that may result in the formation of scar tissue (adhesions, fibrosis) inside the pelvis and other regions of the body [2]. It is related to numerous different factors that contribute to its development. Its origin could be due to retrograde menstruation or to cell metaplasia. It could also be due to the cells outside the uterus, which change into endometrial-like cells and start to grow, or to stem cells. Other factors can also contribute to the growth or persistence of ectopic endometriosis. Endometriosis is known to be dependent on estrogen, which facilitates the inflammation, growth, and pain associated with the disease [2]. Common symptoms include pelvic and abdominal non-menstrual pain, ovulatory pain, pain during urination, dyspareunia, and dyschezia. The symptomatology may present in a mild or severe form, with no correlation of symptom severity to the stage of endometriosis [3]. The underlying pathophysiology of endometriosis remains uncertain; numerous hypotheses have been formulated, but none can fully explain the various clinical presentations of the disease. Thus, endometriosis is believed to have a multifactorial origin that includes histologic, hormonal, and immunologic factors [2].

The diagnosis of endometriosis is exclusively laparoscopic, so the actual prevalence rate in the general population remains unknown. However, it is estimated to afflict 10% of all women based on population prevalence estimates of symptoms [4], which means that 176 million women worldwide suffer from endometriosis [5].

The World Health Organization demonstrated the gravity of endometriosis and showed that it could have negative effects on people’s sexual and reproductive health, quality of life (QoL), and overall well-being [4]. Several studies before the pandemic highlighted that this disease and its complications can have an impact on QoL, affecting physical and psychological health and social and working life [6,7,8,9,10]. The QoL is negatively related to anxiety and depression in women with endometriosis [11]. The relationship between QoC and QoL has often been debated [12]. Chronic pathologies, such as endometriosis, require careful and continuous management. Thus, the quality of care (QoC) and the management of the disease are considered fundamental conditions for the effective prevention of complications and the improvement of the quality of life of patients [13,14].

The spread of the SARS-CoV-2 epidemic in Italy during 2020 forced the National Health Service to interrupt some health services, which were subsequently reorganized based on the implementation of all the prevention measures necessary for the containment of the infection [15,16,17]. The health emergency had a negative impact on the services offered by the National Health System, with the reshaping of deferred scheduled interventions, the postponement of visits and instrumental examinations, and the interruption of prevention and health promotion activities [18,19,20,21]. In this situation, there was a concern for medium and long-term consequences to health, with particular regard to frail people or patients with chronic diseases, such as endometriosis, taking into account the difficulties related to the management and the impact on the quality of life of these diseases [22,23,24,25].

As reported in the literature, patient-oriented care could help the quality life of women with endometriosis, in particular if it guarantees “continuity”, “respect” and “information” [26]. Furthermore, the COVID pandemic had an impact on mental health in the general population and specific groups such as healthcare workers, patients, the elderly, and students [27,28,29,30]. According to our hypothesis, the spread of the epidemic and the changes that took place in the management of healthcare could have had an impact on the quality of care of patients with endometriosis, and consequently on their perceived quality of life. To the best of our knowledge, there were no studies carried out during the pandemic aimed at investigating the quality of life in women with endometriosis and the eventual impact on it or the quality of care, at least not in Italy. This study aims to contribute to the knowledge of this phenomenon by investigating the perceived quality of life of women with endometriosis living in Italy considering the quality of perceived care among the possible predictors of physical and mental quality of life.

## 2. Materials and Methods

This was an online survey carried out from July to September 2021 that was authorized by the Internal Review Board (IRB) of the University of L’Aquila (Protocol number 26/2021).

### 2.1. Participants and Recruiments

The project and the research objectives were presented with an information note disseminated through the support and awareness-raising activity of the administrators of the groups of Italian women with endometriosis registered on social networks such as Facebook and Instagram. Various associations were also contacted, and the dissemination and awareness of participation were also supported by three public figures on Instagram. The inclusion criteria included being a woman, living in Italy, having received a diagnosis of endometriosis, and understanding the Italian language. The exclusion criteria were age < 18 and the lack of informed consent and data processing authorization. Each participant was guaranteed anonymity and respect for privacy.

### 2.2. Tools

A self-reported questionnaire created on the Google Forms web platform was used for the survey, which was developed after a literature review on the quality of life in the population investigated and validated with a group of 55 women with endometriosis (see in Appendix A). The questionnaire included items to provide socio-demographic data, including smoking, alcohol consumption (intake of at least one drink per day) and clinical information about BMI, time of endometriosis diagnosis, pregnancy, treatment, complications related to endometriosis, and comorbidities. It also included the following sections:

Perceived quality of life: For this section, the Italian version of the Health Questionnaire SF-36 V2 Standard was used [31,32]. The SF-36 is a standardized questionnaire that allows one to assess perceived physical and mental health. It is articulated through 36 questions to evaluate 8 subscales (domains): physical functioning (PF), role limitations due to physical health (RF), role limitations due to emotional problems (RE), pain (P), general health (GH), vitality (energy/fatigue: EF), social functioning (SF), and emotional well-being (EW). From SF-36 sub-scales scoring, the following standardized components were calculated: physical component summary (PCS) and mental component summary (MCS). Higher scores indicate a better quality of life [33].

Quality of care received: this was assessed through 12 items: 11 were from a validated questionnaire for assessing the quality of care for people with diabetes [34,35]. Respondents were asked to rate the satisfaction regarding the health care unit that primarily took care of them during the last 12 months concerning the following:Opening hours of the structure;Accessibility of the rooms;Cleanliness and agreeability;Courtesy and helpfulness;Understandable explanations;Being listened to;Waiting times from booking to the visit;Waiting time from arrival to the hospital/clinic;Waiting time from arrival to visit;Overall rating of the service offered in the last 12 months.Judgment on the level of coordination between all services and professionals who deal with the disease.

Each item provided four options: two positives, including “excellent/good”, or “adequate/excellent”, and two less favorable, as, for example “not adequate, just adequate”, or “not sufficient, sufficient”. A final satisfaction index ranging from 0 to 36 was calculated by assigning a score from 0 to 3 to the four options. Higher scores indicated better quality of care.

Women were also asked to report the perception of the impact of the COVID pandemic (neutral/positive/negative) on the quality of health care.

### 2.3. Sample Size

This study involved a convenience sample obtained with the Snowball sampling technique via emails, social media networks, and instant messaging applications. For the estimate of the sample size, referring to a large population (considering the number of members of groups or associations > 30,000), to a precision of ±3% with a 95% confidence interval, to a response level for a single parameter equal to 50%, the estimated sample size was 1032 units [36].

### 2.4. Statistical Analysis

Absolute frequencies and percentages or mean and standard deviations were calculated for each categorical or numerical variable. Comparisons between continuous variables were carried out with a Student’s *t*-test for independent samples or the non-parametric analogue if the normal conditions of the variables were not met. To analyze the factors related to quality of life, we ran two robust regression models for PCS and MCS as dependent variables, and the following independent variables which had been evidenced in other studies, to be related to quality of life in women with endometriosis [37]: age, occupational status (yes/no), educational level (high/how), BMI, physical activity (yes/no), smoker (yes/no), alcohol consumption (yes/no), full-term pregnancy (yes/no), surgical treatment (yes/no), hormonal treatments-lifetime perspective (yes/no), comorbidities (yes/no), clinical complications (yes/no), and satisfaction index, which was investigated as a new variable in this article, as reported in Figure 1).

Variables entered into regression models were collected on women who had had a medical visit for endometriosis in the 12 months prior to the interview. Data on pregnant women were excluded from the analyses. We first performed a univariate analysis to explore the separate effect of a variable on the summary components PCS and MCS, then two multivariable robust regression models for PCS and MCS were run adjusting for all factors or covariates investigated. All analyses were performed using STATA 14 software [38].

## 3. Results

One thousand sixty-five women with a self-reported diagnosis of endometriosis participated in the survey. The characteristics of the sample are shown in Table 1. The average age of the participants was 35 years (SD: 8). Only 24 women were of foreign nationality, and 62% indicated that they were married or cohabiting. A total 11% have a low level of education (elementary/middle school), 45% have a high school diploma, and about 44% have a degree and post-graduate degree. A total of 304 reported having completed one or more pregnancies (29%), about 47% were diagnosed with endometriosis between the ages of 20–29, and in 35% of cases the time elapsed between the onset of symptoms and diagnosis was at least 10 years. Some 61% (649) underwent surgery for endometriosis, and most reported having complications from it (*n* = 824: 77%).

We found that the highest score reported by women interviewed was for physical functioning, with a mean of 66.65 (+/−27.12), while the mean level of vitality was the lowest: 35.70 (+/−18.40). The mean levels of PCS and MCS were 38.89 (+/−10.55) and 34.59 (+/−11.17), respectively (Table 2).

One hundred and seventy-eight women out of 1065 interviewed reported that they had not received any kind of medical care in the last 12 months, while 889 of the participants had a medical visit during the 12 months prior to the study; most of them (710/889: 80%) were treated in a specialized center for endometriosis, while the remaining 20% visited a gynecologist or family doctor.

As reported in Table 3, over 80% of the subsample of 889 participants who had a medical visit during the previous 12 months expressed positive evaluations (good or excellent) with respect to the opening hours of the facilities (approximately 86%), accessibility of the premises (approximately 95%) cleanliness and pleasantness (94%), courtesy and availability (85%), understanding (about 81%). A total of 73% of women reported being listened to often and always. In general, 77% of women judged the services received positively (good or excellent), and 51% considered the coordination between the professionals involved in the care to be satisfactory (good or excellent). The global satisfaction index mean was 23.11 (4.8), with a median of 24 and a range from 5 to 34.

Considering the negative perceptions of quality of the first eight dimensions investigated, as indicated in the previous table, due to the COVID pandemic, most of the participants (41%) reported the waiting times from booking to the visit were lengthened, and it was the dimension most affected by the pandemic; in contrast, the least affected was cleanliness and agreeability (2%).

As reported in Table 4, a multivariable regression analysis showed that PCS was positively related to occupation (beta = 2.301; *p* = 0.002), high educational level (beta = 3.206; *p* < 0.001), physical activity (beta = 2.030; *p* = 0.003), and health care satisfaction (beta = 0.382; *p* < 0.001). The quality of physical health in smokers, in alcohol consumers, and in women with children and other diseases was lower. In fact, PCS was negatively associated with smoking (beta = −2.286; *p* = 0.006), alcohol use (beta = 2.038; *p* = 0.009), full- term pregnancy (beta = −2.513; *p* = 0.003), and comorbidities (beta = −5.035; *p* < 0.001), (Table 4).

As reported from the multivariable regression analysis, factors such as age, physical activity, smoking, comorbidities, and health care satisfaction were significantly associated with quality of mental health. Quality of mental health was higher in older women (beta = 0.151; *p* = 0.012), in women who practiced physical activity (beta = 1.867; *p* = 0.018), and in women satisfied with their health care (beta = 0.438; *p* < 0.001). Women who smoked and women who reported comorbidities had the lowest level of MCS (beta= −2.298, *p* = 0.016; and beta= −1.645, *p* = 0.036 respectively) (Table 5).

## 4. Discussion

This was an online cross-sectional survey which included standardized measures, and 1065 records were collected at the closing date of the recruitment process. The geographical distribution of responses was 40% from Northern Italy, 22% from Central Italy and 38% from Southern Italy and the Islands. It was compatible to the distribution of the Italian female population in the three territorial areas: the North, Center, South, and the Islands, according to the Italian National Institute of Statistics (ISTAT) [39]. To the best of our knowledge, this was the first study to investigate the quality of life among women with endometriosis during the COVID pandemic in Italy.

It is known that the quality of life in women with endometriosis is severely tested because the disease impacts physical and mental health and, consequently, social life [40].

Some research has shown that endometriosis, whether symptomatic or pain-free, reduces quality of life, work productivity, and mental health (e.g., anxiety and depression) [10,41,42] because it is characterized by uncertainty about the course of the disease and the future in general, with pervasive concerns about crucial aspects of a woman’s life, such as sexuality and infertility [43]. Endometriosis-related mental distress may be correlated with increasing age [43,44].

Our results showed that 85% of participants were diagnosed with endometriosis between the ages of 20–39 years. The mean diagnostic delay was estimated to be 7 years (SD = 3): most of the interviewees (57%) reported having been diagnosed at least 5 years after the onset of symptoms, and 367 women (35%) were diagnosed after 10 years or more. These results are in line with the Italian national average [45].

Diagnostic delay can result in increased symptoms and illness severity, the worsening of physical and psychological sequelae, and delayed access to adequate treatment and care, all of which lead to higher healthcare consumption and expenditures [46,47,48].

Diagnostic delay is usually a factor in chronic illnesses, and it is especially common in endometriosis since the diagnosis is dependent on histological confirmation. Furthermore, endometriosis-related stigma is an emerging factor associated with diagnostic delay, and a lack of endometriosis awareness among women, their families, and intimate partners, as well as health care providers and the general population, perpetuates stigma and its negative effects on health and psychosocial well-being [49].

A total of 64% of participants (677/1065) were treated with hormonal therapies. This finding could be related to the recommendations by the European Society of Human Reproduction and Embryology (ESHRE). Medical therapy should be recommended for women with symptomatic endometriosis and should be aimed at controlling pain and preventing the progression of existing lesions. The most common pharmacological treatment is the hormonal type, mainly based on estrogen-progestogen pills, progestin pills or GNRH analogues [50].

In this study, 716 (67.2%) women had dyspareunia, and 672 (63.1%) had chronic pelvic pain. These are the most frequently reported symptoms among women with endometriosis [51,52], and previous studies reported that dyspareunia and pain negatively affect sexual life and quality of life in general in women with endometriosis [40,53]. In fact, it is no coincidence that in this survey the perception of quality of life, the domain linked to the perception of one’s own state of health, was the lowest (mean = 37.18, SD = 20.77), together with the domain of vitality (energy/fatigue), with an average of 35.70 (SD = 18.40). Our sample perceived level of physical health quality to be higher than mental health: PCS = 38.89 (SD = 10.55) and MCS = 34.59 (SD = 11.17). These results are in line with data reported by He G and colleagues, even if they were higher if calculated in all samples (PCS: 42.0 ± 8.99; MCS: 44.0 ± 11.91), and lower in women with anxiety: PCS = 38.4 (8.37) and MCS = 36.4 (11.55) [11].

However, our results are in contrast with the findings reported in a pre-pandemic cross-national study carried-out in Poland by Agnieszka Bień (psychological domain:13.33 ± 2.28, physical domain: 11.52 ± 3.02), where even the QoL was measured with different tools [54], and this could be related to the negative impact that the COVID pandemic had on the population. In particular, a recent study on women with endometriosis during the pandemic highlighted that patients with endometriosis were at a high risk of developing mental symptoms [55], demonstrating the importance of social connections for wellbeing and life satisfaction reported in the pre-pandemic era [56].

To introduce the quality of health care perception as a factor related to QoL, we asked about quality of care to women treated during the previous 12 months, and we had a subsample of 899 women. A total of 106 women (16%) had not had a medical visit for endometriosis during the previous 12 months, and this is an important finding of this study that warns of the need to monitor the health status in this particular patient group. Endometriosis is typically a progressive condition, meaning it can get worse over time [57]; therefore, regular checks are essential to ensure better treatment.

Among women who reported a medical visit, 710 (80%) were treated at an endometriosis center, and 20% were primarily treated by gynecologists and general doctors. Most interviewees (>80%) appeared satisfied with the health services and care they got during the first 12 months of the pandemic. This is a good result considering the effort made to maintain the continuity of health services during the pandemic. Disappointment was reported by only 6% of women (55/889), and it mainly concerned the following aspects: the times of the appointments of the structures (5%: 44/889), the courtesy of the staff (5.5%: 49/889), the patients’ willingness to listen (5%: 44/889), and above all the coordination between the various professional health workers (22%: 195/889).

The multivariable regression analysis showed that occupation (beta = 2.446; *p* = 0.001) and high educational level (beta = 3.418; *p* < 0.001) are related to a higher PCS. Despite the universal nature of the Italian NHS, a diagnosis of endometriosis probably has a direct and indirect impact on the economic situation of the patients and their families [58], since women often need to move to specialized centers that are far from their place of residence, and this could be related to the problem of inequalities in health care [59,60,61]. Our results are in line with studies by other authors showing that women with endometriosis with a higher level of education have a better quality of life in the physical domain) [54,62]. In addition, according to Bień, family wealth and financial resources are important factors in the treatment of endometriosis and its complications, and are related to QoL [54].

The quality of physical health improves with physical activities (beta = 2.030; *p* = 0.003) and health care satisfaction (beta = 0.382; *p* < 0.001), as reported above. In their study, Goncalves et al. [63] reported that the degree of daily pain was significantly lower in the group of women who engaged in physical activity. However, a previous meta-analysis [64] found that the relationship between physical activity and endometriosis has been inconsistent, and this aspect should be investigated more comprehensively [65].

According to our findings, in the presence of children, in the presence of other diseases, and in women who smoke, physical health is experienced with less intensity. In fact, PCS was significantly negatively associated with smoking (beta = −2.286; *p* = 0.006), full-term pregnancy (beta = −2.513; *p* = 0.003), and comorbidities (beta = −5.035; *p* < 0.001). These results are similar to the findings from a recent cohort study among 1091 women that reported that currently smoking and comorbidities were factors associated with the decline in the PCS score of the Short Form 36 [66]. Another important finding of the study was the negative relation between PCS and alcohol consumption. There was evidence to indicate a relationship between alcohol consumption and endometriosis risk [67].

With respect to mental health, we found that age, physical activity and a good perception of health care were predictors of higher levels of mental health (age: beta = 0.151; *p* = 0.012; physical activity: beta = 1.864; *p* = 0.018; health care satisfaction: beta = 0.435; *p* < 0.001). Women who smoke and women who reported comorbidities had the lowest level of MCS (beta = −2.345; *p* = 0.013 and beta = −1.63; *p* = 0.37 respectively). Our findings are consistent with the narrative review by Capezzuoli et al. which showed how gynecological and systemic comorbidities may negatively affect the quality of life and the global health of women with endometriosis [68]. In addition, several studies indicate a strong relationship between mental health disorders (depression, anxiety disorders, substance abuse, panic, somatoform disorders, etc.) and endometriosis, with a great impact on the mental quality of life of these women [11,69].

This study has several limitations. First of all, it was a convenience sample, the questionnaire was self-completed online and then the selection bias could be met, taking into account that internet access is also related to age; in fact, according to ISTAT data, internet use is limited in women over the age of 65, especially those with a low level of education [70]. Another limitation is the study design (transversal study), that did not allow for the establishment of causal relationships [71]. We do not have information on menopausal women, and this is an important factor to analyze, in addition to information about sexual problems related to endometriosis or the self-management of it [72]. Despite the present limitations, the study aims to contribute to the knowledge of the phenomenon and to raise awareness of women with endometriosis in the difficult period of the pandemic. Further studies are needed to further investigate the long-term issues that result from the pandemic, also considering other aspects that have not been investigated in this study, such as the distinction between menopausal women and non-menopausal women.

## 5. Conclusions

The COVID pandemic had a negative impact on the QoL of the general population, mostly because of social restrictions, but also because people affected by chronic illnesses—such as endometriosis—experienced this lowering of the QoL to a greater extent because of the worse quality of health care received.

It is known how endometriosis negatively impacts quality of life and social life [73]. The main finding of our study is that the positive perception of quality of healthcare is a significant predictor of a good quality of life, both physically and mentally. The satisfaction with health care was found as a significant predictor of QoL in women with endometriosis, both for physical and mental health, although the study reported a slight influence on mental health and a stronger influence on physical health. This result is very important in maintaining attention on the need to develop specialized pathways to help women with endometriosis and for early diagnosis, especially in the era of the pandemic, in which the worst problems reported by the women of the interviewed sample are related to being on a waiting list.

## Figures and Tables

**Figure 1 ijerph-20-00625-f001:**
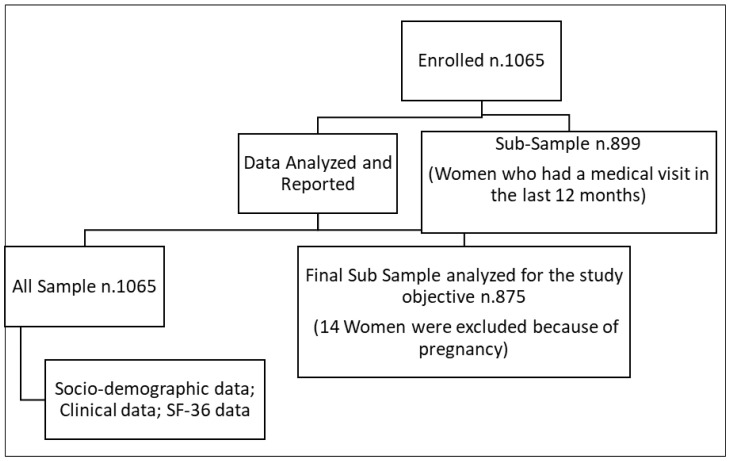
Flow chart diagram.

**Table 1 ijerph-20-00625-t001:** Sample characteristics: socio-demographic and clinical data (All participants *n*. 1065).

Characteristics	Mean (SD) or N (%)
Age ^a^	35 (8)
Residence	
South	408 (38%)
Center	229 (22%)
North	428 (40%)
Nationality	
Italian	1041 (98%)
Foreign	24 (2%)
Marital status ^a^	
Married/cohabiting	629 (59%)
Single	436 (41)
Living alone	
No	950 (89%)
Yes	115 (11%)
Education ^a^	
High (degree or above)	467 (44%)
Low (secondary school)	598 (56%)
Employed ^a^	
Yes	696 (65%)
No	369 (35%)
Body Mass Index ^a^	23.11 (4.8)
Physical activity ^a^	
Yes	435 (41%)
No	630 (59%)
Smoking ^b^	
Tobacco	221 (21%)
E-cig	60 (5%)
No	784 (74%)
Alcohol consumption ^a^	
Yes	265 (25%)
No	780(75%)
Full-term pregnancies ^a^	
Yes	304 (29%)
No	761 (71)
Time from symptoms to diagnosis (years)	
<1	203 (19%)
1–4	253 (24%)
5–9	236 (22%)
≥10	367 (35%)
Diagnostic delay (years)	7 (3.7)
Hormonal treatments (lifetime perspective) ^a^	
Yes	677 (64%)
No	388 (36%)
Clinical complications related to endometriosis ^c^	824 (77%)
Chronic pelvic pain	672 (63.1%)
Dyspareunia	716 (67.2%)
Pelvic floor disorders	525 (49.3%)
Self-catheterisms	38 (3.6%)
Neuropathy/nerve disorders	331 (31.1%)
Infertility	378 (35.5%)
Hysterectomy	108 (10.1%)
Salpingectomy	140 (13.2%)
Ovariectomy	106 (9.9%)
Intestinal stenosis	123 (11.6%)
Intestinal resection	157 (14.7%)
Intestinal stoma/urostomy	55 (5.2%)
Bladder resection	69 (6.5%)
Adhesions	592 (55.6%)
Surgical intervention for endometriosis ^a^	
No	416 (39%)
Yes	649 (61%)
Comorbidities ^a^	
No	487 (46%)
Yes	578 (54%)

^a^: Factors included in the physical and mental component score (PCS-MCS) regression; ^b^: Factors included in the physical and mental component score (PCS-MCS) regression excluding smoking electonic cigarettes; ^c^: Factors included in the physical and mental component score (PCS-MCS) regression as dichotomic (yes/no).

**Table 2 ijerph-20-00625-t002:** SF36 Scale scores and summary components (All participants *n* = 1065).

SF-36 Scale Scores	Mean (SD)
Physical functioning	66.65 (27.12)
Role limitations due to physical health	39.50 (38.92)
Role limitations due to emotional problems	40.56 (41.14)
Pain	46.71 (27.92)
General health	37.18 (20.77)
Vitality (Energy/fatigue)	35.70 (18.40)
Social functioning	44.47 (25.61)
Emotional well-being	47.88 (19.24)
**SF-36 Component Scores**	**Mean (SD)**
Physical component summary (PCS)	38.89 (10.55)
Mental component summary (MCS)	34.59 (11.17)

**Table 3 ijerph-20-00625-t003:** Quality health care perception (women who had a medical visit for endometriosis in the last 12 months, *n* = 889).

Items	N (%) or Mean (SD)/Median (Range)
Opening hours of the structure	
Not adequate	44 (5%)
Just adequate	82 (9.2%)
Adequate	520 (58.5%)
Excellent	243 (27.3%)
Accessibility of the rooms	
Not very accessible	14 (1.6%)
Accessibility with some difficulty	31 (3.5%)
Fairly accessible	353 (39.7%)
Very accessible	491 (55.2%)
Cleanliness and agreeability	
Not sufficient	8 (0.9%)
Sufficient	41 (4.6%)
Good	303 (34.1%)
Optimum	537 (60.4%)
Courtesy and helpfulness	
Not sufficient	49 (5.5%)
Sufficient	81 (9.1%)
Good	241 (27.1%)
Optimum	518 (58.3%)
Explanations understandable	
Never	19 (2.2%)
Sometimes	151 (17%)
Always	468 (52.6%)
Often	251 (28.2%)
Being listened to	
Never	44 (5%)
Sometimes	193 (21.7%)
Always	258 (29%)
Often	394 (44.3%)
Waiting times from booking to the visit	
Maximum one month	428 (48%)
Between 1 and 6 months	357 (40%)
Between 6 and 12 months	88 (10%)
Over a year	16 (2%)
Waiting time from arrival to visit	
Less than 15 min	259 (29%)
Between 15 min and 30 min	337 (38%)
Between 30 min and an hour	169 (19%)
More than an hour	124 (14%)
Overall rating of the service offered in the last 12 months	
Not sufficient	55 (6%)
Sufficient	149 (17%)
Good	419 (47%)
Optimum	266 (30%)
Judgment of the level of coordination between all services and professionals who deal with the disease	
Not sufficient	195 (22%)
Sufficient	207 (23%)
Good	305 (34%)
Optimum	182 (21%)
Preventive anti-COVID measures	
Not sufficient	17 (1.9%)
Sufficient	74 (8.3%)
Good	319 (35.9%)
Optimum	479 (53.9%)
Global satisfaction index ^a^	23.11 (4.8)/24 (5–34)

^a^: Factors included in the physical and mental component score (PCS-MCS) regression.

**Table 4 ijerph-20-00625-t004:** Predictors of high level of PCS (*n* = 875 *).

	Univariate Regression			Multivariable Regression		
	Beta	S.E.	*p*-Value	Beta	S.E.	*p*-Value
Age	−0.186	0.044	<0.001	−0.094	0.053	0.073
Occupational status: employed	3.449	0.721	<0.001	2.301	0.750	0.002
Educational level: high	5.180	0.681	<0.001	3.206	0.717	<0.001
BMI	−0.265	0.072	<0.001	−0.070	0.073	0.343
Physical activity: yes	3.647	0.698	<0.001	2.030	0.687	0.003
Smoke: yes	−3.121	0.854	<0.001	−2.286	0.828	0.006
Alcol: yes	3.358	0.793	<0.001	2.038	0.779	0.009
Full term pregnancy: yes	−4.465	0.754	<0.001	−2.513	0.845	0.003
Surgery: yes	−1.563	0.713	0.029	−0.741	0.728	0.309
Therapy: yes	1.132	0.729	0.121	0.135	0.737	0.854
comorbidities: yes	−6.680	0.662	<0.001	−5.035	0.682	<0.001
Complications: yes	0.374	0.833	0.653	0.140	0.810	0.863
Satisfaction index	0.401	0.668	<0.001	0.382	0.061	<0.001

* Women were pregnant at the moment of the survey were excluded from this analysis; the beta are the regression coefficients.

**Table 5 ijerph-20-00625-t005:** Predictors of high level of MCS (*n* = 875 *).

	Univariate Regression			Multivariable Regression		
	Beta	S.E.	*p*-Value	Beta	S.E.	*p*-Value
Age	0.278	0.046	<0.001	0.151	0.060	0.013
Occupational status: employed	2.929	0.754	<0.001	1.699	0.861	0.049
Educational level: high	−1.272	0.734	0.083	−1.451	0.823	0.078
BMI	0.112	0.075	0.137	−0.027	0.084	0.746
Physical activity: yes	2.362	0.736	<0.001	1.867	0.789	0.018
Smoke: yes	−2.205	0.891	0.013	−2.298	0.949	0.016
Alcohol: yes	−0.169	0.837	0.840	−0.444	0.894	0.620
Full term pregnancy: yes	2.915	0.798	<0.001	0.705	0.970	0.467
Surgery: yes	2.844	0.740	<0.001	0.791	0.836	0.344
Therapy: yes	−2.541	0.757	0.001	−1.530	0.846	0.071
Comorbidities: yes	−1.227	0.730	0.093	−1.645	0.782	0.036
Complications: yes	−0.208	0.870	0.811	−0.109	0.930	0.906
Satisfaction index	0.480	0.070	<0.001	0.438	0.070	<0.001

* Women who were pregnant at the moment of the survey were excluded from this analysis; the beta are the regression coefficients.

## Data Availability

The data analyzed in this study are not available to outside researchers due to privacy issues.

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
