# Peer review of "Does Quality of Care (QoC) Perception Influence the Quality of Life (QoL) in Women with Endometriosis? Results from an Italian Nationwide Survey during Covid Pandemic"

_ijerph, 2022, doi:10.3390/ijerph20010625_

Round 1

Reviewer 1 Report

Manuscript ID: ijerph-2089698

Title: Does quality of care (QoC) perception influence the quality of life (QoL) in women with endometriosis? Results from an Italian nationwide survey during Covid pandemic.

This study aimed to investigate the quality of life in women with endometriosis and the eventual impact on it or the quality of care during the COVID-19 pandemic. The topic described in this manuscript is interesting. This is a valuable submission that I recommend for publication.

Author Response

We thank the Reviewer for the time and effort to revise the manuscript. Thank you for your kind words. We sincerely appreciate  your positive comment.

Reviewer 2 Report

The inclusion criteria included being a woman, living in Italy, having received a diagnosis of endometriosis, and understanding the Italian language. The exclusion criteria were age <18 and the lack of informed consent and data processing authorization. Each participant was guaranteed anonymity and respect for privacy. I think that for your further research it would be appropriate to add the duration of the diagnosis and stage of the disease to the criteria to determine the quality of life.

Author Response

We thank the Reviewer for the time and attention to revise the manuscript We sincerely appreciate your valuable comment and suggestion, which will help us in improving the quality of further researchs.

Reviewer 3 Report

The article is well written and easy to understand at first reading by health professionals as well as others.

There’s no noticeable significant bias in the way the study was planned and the statistics were performed, nor in the conclusions drawn by the authors.

The primary goal of the study, which is proving the link between quality of care (QoC) perception and quality of life (QoL) in women with endometriosis, is very relevant because it can also apply to other chronic illnesses and might justify a greater investment in the medical care.

Given the main finding of the study, which is that the perception of quality of healthcare is a significant predictor of both physical and mental QoL, it can also be stated what follows: Covid pandemic had a negative impact on the QoL of the general population, mostly because of the social limitations, but people affected by chronic illnesses - like endometriosis - experienced this lowering of the QoL to a greater extent because of the worse quality of health care received.

Limits of the study are honestly recognised and reported by the authors (lines 327-339).
The number of participants could have probably easily been greater, e.g. by carrying out the online survey for a longer time than just 2 months; all the more reason because by excluding the women who hadn’t had a medical visit for endometriosis in the 12 months prior to the interview the sample size sinks to 899 (875 by excluding the pregnant ones), making the final sub-sample smaller than the initially estimated convenience sample size which was 1032 units.

Author Response

The primary goal of the study, which is proving the link between quality of care (QoC) perception and quality of life (QoL) in women with endometriosis, is very relevant because it can also apply to other chronic illnesses and might justify a greater investment in the medical care.

We thank the Reviewer for the time and effort to revise the manuscript.

Thank you very much for agreeing with us to the intention of this manuscript. We have read your comments carefully and tried our best to address them one by one.

Given the main finding of the study, which is that the perception of quality of healthcare is a significant predictor of both physical and mental QoL, it can also be stated what follows: Covid pandemic had a negative impact on the QoL of the general population, mostly because of the social limitations, but people affected by chronic illnesses - like endometriosis - experienced this lowering of the QoL to a greater extent because of the worse quality of health care received.

We thank the reviewer for the kind suggestion. We revised the manuscript reporting the suggested sentence.

Limits of the study are honestly recognised and reported by the authors (lines 327-339). The number of participants could have probably easily been greater, e.g. by carrying out the online survey for a longer time than just 2 months; all the more reason because by excluding the women who hadn’t had a medical visit for endometriosis in the 12 months prior to the interview the sample size sinks to 899 (875 by excluding the pregnant ones), making the final sub-sample smaller than the initially estimated convenience sample size which was 1032 units.

Thank you for your important comment. We will consider your suggestion for the next study, and we will lengthen the time for the survey.

Reviewer 4 Report

The authors of the proposed manuscript investigated the quality of life among women with endometriosis during the Covid-19 pandemic in Italy. The standard of the use of English is acceptable, and tables and all the figures are present. Experimental design - The research is original and creative.

Validity of the findings - Analysis and conclusions are supported by the data.

In the discussion section, the literature is nicely cited.

In the introduction part, the information about endometriosis is shortly described. This paragraph can be extended, but it is not mandatory. 

The invitation/survey was sent to participants via social media (Meta) and Google Forms (Google).  One thousand sixty-five women with a self-reported diagnosis of endometriosis participated in the study/survey.

Please add to your manuscript the questionnaire in Italian and English. This helps the next group to replicate your study out of Italy. Please add as a supplementary material.

In the results part, the authors wrote, “Only 24 women are of foreign nationality” I could not find this information in table no. 1.

Line 175. “l” is missing in the word medical.

According to Bien’s study [reference no. 52], the authors wrote, “have to pay for medication, fertility treatment and complementary therapies (acupuncture, yoga, physiotherapy, psychotherapy), and even sanitary towels”. Did You observe the same?

Others:

In reference no. 50, vanish for the first time doi number, which is not present in the rest. Please standardize and use one style. This same is observed in 65, 66… and so on.

In reference no. 54, the number 54 is double.

In reference no. 72, please use the new line.

Author Response

The authors of the proposed manuscript investigated the quality of life among women with endometriosis during the Covid-19 pandemic in Italy. The standard of the use of English is acceptable, and tables and all the figures are present. Experimental design - The research is original and creative.

Validity of the findings - Analysis and conclusions are supported by the data.

In the discussion section, the literature is nicely cited.

We thank the Reviewer for the time and effort to revise the manuscript. Thank you very much for your positive evaluation.

We read your comments carefully and tried our best to address them one by one

In the introduction part, the information about endometriosis is shortly described. This paragraph can be extended, but it is not mandatory. 

Thank you so much for your comment. We have extended the paragraph during the revision of the manuscript.

The invitation/survey was sent to participants via social media (Meta) and Google Forms (Google).  One thousand sixty-five women with a self-reported diagnosis of endometriosis participated in the study/survey.

Please add to your manuscript the questionnaire in Italian and English. This helps the next group to replicate your study out of Italy. Please add as a supplementary material.

We added the questionnaire ( Italian and English version) as a supplementary material.

In the results part, the authors wrote, “Only 24 women are of foreign nationality” I could not find this information in table no. 1.

Thank you so much for your concern. We added “nationality” in table 1. We are sorry for not reporting it before

Line 175. “l” is missing in the word medical.

Thank you for your attention. We revised it .

According to Bien’s study [reference no. 52], the authors wrote, “have to pay for medication, fertility treatment and complementary therapies (acupuncture, yoga, physiotherapy, psychotherapy), and even sanitary towels”. Did You observe the same?

In our study even though we did not ask participants for income or for Perceived family wealth as reported by Bien, we have investigated the availability of a job that was significantly related to a better QoL perception. For this finding we agree with Bien that the economic aspect is very important for the QoL in women with endometriosis.  Our questionnaire included an open question in which we asked women what they proposed for improve their QoL (open question that we intend to analyze with specific software) :  most of the reported proposals concern the possibility of economic aid or agreements to deal with the various costs of the disease, due also to complementary therapies (physiotherapist for example). Therefore, even if the present manuscript doesn't report this results because  the lack of analysis of the specific item, we agree with Bien that the economic problem is linked to QoL as in general in chronic diseases.

We are developing another project to understand the costs associated with endometriosis and the impact (social and economic impact) of the absence from work for women with endometriosis in Italy.

Others:

In reference no. 50, vanish for the first time doi number, which is not present in the rest. Please standardize and use one style. This same is observed in 65, 66… and so on.

In reference no. 54, the number 54 is double.

In reference no. 72, please use the new line.

Thank you so much for your suggestions. We revised it.
